# Learning Retrieval Augmentation for Personalized Dialogue Generation

**Qiushi Huang**[1,2]**, Shuai Fu**[2]**, Xubo Liu**[1]**,Wenwu Wang**[1]**,**
**Tom Ko**[3]**, Yu Zhang**[2*]**, Lilian Tang**[1*]

[1]University of Surrey, [2]Southern University of Science and Technology, [3]ByteDance AI Lab
{qiushi.huang, xubo.liu, w.wang, h.tang}@surrey.ac.uk,
{fus.akhasi, tomkocse, yu.zhang.ust}@gmail.com

## Abstract

Personalized dialogue generation, focusing on generating highly tailored responses by leveraging persona profiles and dialogue context, has gained significant attention in conversational AI applications. However, persona profiles, a prevalent setting in current personalized dialogue datasets, typically composed of merely four to five sentences, may not offer comprehensive descriptions of the persona about the agent, posing a challenge to generate truly personalized dialogues. To handle this problem, we propose **L**earning Retrieval **A**ugmentation for **P**ersonalized **D**ial**O**gue **G**eneration (**LAPDOG**), which studies the potential of leveraging external knowledge for persona dialogue generation. Specifically, the proposed LAPDOG model consists of a story retriever and a dialogue generator. The story retriever uses a given persona profile as queries to retrieve relevant information from the story document, which serves as a supplementary context to augment the persona profile. The dialogue generator utilizes both the dialogue history and the augmented persona profile to generate personalized responses. For optimization, we adopt a joint training framework that collaboratively learns the story retriever and dialogue generator, where the story retriever is optimized towards desired ultimate metrics (e.g., BLEU) to retrieve content for the dialogue generator to generate personalized responses. Experiments conducted on the CONVAI2 dataset with ROCStory as a supplementary data source show that the proposed LAPDOG method substantially outperforms the baselines, indicating the effectiveness of the proposed method. The LAPDOG model code is publicly available for further exploration. [1]

## 1 Introduction

Personalized dialogue generation (Zhang et al., 2018; Dinan et al., 2019), which prompts an agent

---

*Corresponding authors.
[1]https://github.com/hqsiswiliam/LAPDOG

to generate consistent responses based on historical dialogue context and given persona profiles, has recently drawn substantial attention in many applications. For instance, such an agent could effectively adapt to different roles such as a customer service representative by tailoring its responses to specific customer needs based on its persona and improving customer interaction and satisfaction. Besides, personalized responses can foster a sense of human-like interaction in social platforms, thereby enriching the user experience.

The persona profiles contain background sentences describing the agent (e.g., *I like to go hunting.*) and play a crucial role in customizing the dialogue. Ideally, a persona profile should be as comprehensive as possible, containing diverse and detailed descriptions of an agent. However, these persona profiles, typically consisting of only four to five sentences, do not provide comprehensive descriptions for the persona of the agent. Such lack of depth and diversity in the persona descriptions impedes existing methods (Liu et al., 2020; Song et al., 2021; Huang et al., 2022) from generating highly personalized and contextually rich responses, though they have shown capabilities in producing grammatically correct and human-like responses. In essence, those models are restricted by the static and limited persona profile. Hence, those models fail to dynamically incorporate more intensive extra personalized profiles when decoding the responses.

Though the given persona profile is limited, there are many external textual resources to describe personality and daily life circumstances. Hence, it is intuitive to ask: *can we use other related datasets to enrich the details of the persona profile?* This key question has not been thoroughly explored in existing methods, which primarily rely on the persona profile and dialogue context alone. An immediate issue is which types of external datasets could be used. A promising source is story data since they

encompass diverse life events, personality traits, motivations, and experiences, which can contribute to a more detailed and realistic persona. For example, given the persona sentence as "*I like to work on vintage cars.*", potential retrieved stories' titles can be "*Antique Car Show*" and "*Mechanic*", the details of the story content can be found in the appendix (Table 7). Furthermore, the clear and inherent structure in stories can enhance the consistency of the persona. In this work, we choose story data to facilitate the generation of more engaging and contextually meaningful dialogues.

Given the external knowledge (e.g., story data), how to infuse it into the process of personalized dialogue generation straightforwardly remains challenging. The first hurdle is the lack of explicit annotations for retrieval, which are the key to selecting relevant and helpful content to augment persona profiles. In addition, the criterion for assessing the efficacy of these contents remains unclear. For instance, retrieval-augmented generation (RAG) (Lewis et al., 2020) is based on predicted probability distribution, which may not directly align with the objective of generating personalized responses. Moreover, simply tuning dense retriever (Karpukhin et al., 2020) in a RAG's paradigm may result in suboptimal retrieval outcomes as the retriever is inclined to consistently select similar passages for all queries, which may impede the further expansion of the persona profile details.

In this paper, we give the first try to utilize the story data as external knowledge for the personalized dialogue generation task and propose a **L**earning **R**etrieval **A**ugmentation for **P**ersonalized **D**ial**O**gue **G**eneration (**LAPDOG**) framework. Specifically, the proposed model LAPDOG, consisting of a retriever to retrieve helpful information to enrich the persona and a generator to generate dialogues, is an end-to-end learnable retrieval methodology for integrating additional contextual information into personalized dialogue generation. LAPDOG utilizes non-differentiable metrics (e.g., BLEU, F1, and ROUGE-L) to guide the training of the retriever by aligning the retriever scores to these desired metrics, thereby facilitating the generation of relevant and diverse personalized responses. To ensure diversity in the retrieval process, we design a retrieval candidate augmentation during training, which prevents consistently selecting similar passages for all queries and provides a broader range of contextual inputs for the dialogue generator. In addition to the retrieved content, the persona information and dialogue context are also integrated into the dialogue generator. Furthermore, LAPDOG adopts a cooperative framework wherein the retriever and generator are jointly trained. This process enables LAPDOG to generate personalized responses that are coherent, contextually rich, and in line with the persona of the agent. Unlike other retrieval models (Zhou et al., 2022; Santhanam et al., 2022) that rely on annotated retrieval datasets, our method retrieves the supplementary context in an end-to-end, unsupervised manner, which can be seamlessly extended to other suitable text sources.

We conduct experiments on the CONVAI2 dataset (Dinan et al., 2019), which is widely recognized and extensively studied in the field of personalized dialogue generation (Huang et al., 2022; Song et al., 2021; Liu et al., 2020), and the ROCStory dataset (Mostafazadeh et al., 2016) acts as external knowledge. Experiments demonstrate the positive impact of learnable retrieval augmentation on performance. Quantitatively, the proposed LAPDOG method consistently yields improvements over the baseline models with varying model sizes. Moreover, the retrieved contents offer insights into the rationale behind the generation ability of the generator. Comprehensive ablation studies demonstrate that joint objective guidance outperforms each individual objective and provides insights into the size of retrieval candidates and the use of different metrics.

Overall, our contributions can be summarized as follows.

- We present a novel LAPDOG model for personalized dialogue generation to retrieve relevant contents in external knowledge to the persona using the non-differentiable objective.

- We introduce candidate augmentation as a means to enhance learning retrieval augmentation, resulting in improved performance and increased diversity of candidate selections during the inference process.

- The proposed LAPDOG framework significantly enhances the performance over baselines, showing promising potential for learnable retrieval augmentation on personalized dialogue generation. Our code and pre-trained model will be open-sourced.

## 2 Related Work

### 2.1 Personalized Dialogue Generation

Based on the PersonaChat dataset (Zhang et al., 2018), Dinan et al. curate the CONVAI2 dataset, which contains a brief persona with four to five sentences for each interlocutor. This unique dataset has become a standard benchmark for the personalized dialogue generation task and built on this dataset, there are numerous studies, each of which approaches personalized dialogue generation from diverse perspectives. For example, Wolf et al. proposes a fine-tuning model based on the GPT2 model (Radford et al., 2019). Song et al. integrates three BERT models (Devlin et al., 2019) via reinforcement learning to generate responses. Liu et al. propose a transmitter and receiver model, which utilizes reinforcement learning with manually designed rewards for further refinement, for the personalized dialogue generation task. Cao et al. adopt model-agnostic data augmentation to use language models, such as GPT2 and BERT (Devlin et al., 2019), to augment the training set with pseudo training data. Huang et al. devise an adaptive attention mechanism to integrate information from persona and context encoders seamlessly. In contrast to the aforementioned models, the proposed LAPDOG method introduces an end-to-end dense retriever framework to simultaneously augment the input of the generator from external data source and tune the retriever.

### 2.2 Retrieval-Augmented Text Generation

There are some works to incorporate retrievers into their respective models via different integration strategies to enhance text generation tasks. For instance, DocPrompting (Zhou et al., 2022) curated a retrieval annotation dataset to train a retriever to retrieve and do input augmentation for code generation. Toolformer (Schick et al., 2023) bootstraps retrieval-annotated data, which performs fine-tuning on language models for the retrieval-augmentation ability. FLARE (Jiang et al., 2023) extends the Toolformer to large language models such as (Ouyang et al., 2022) with special-designed prompts. RePlug (Shi et al., 2023) further refines the retriever by distilling the knowledge from the language model's probability. RAG (Lewis et al., 2020) jointly trains the retriever and language model, which updates the retriever by the language model's probability. Different from those models, the proposed LAPDOG model is designed

specifically for personalized dialogue generation with a focus on optimizing desired objectives rather than the language model's probability distribution. Since RePlug, Toolformer, and FLARE are based on large language models or their API calls, we do not include them in the comparison to LAPDOG. Compared with other models, we do not rely on retrieval annotations or bootstrapped retrieval annotations. The training objectives are directly computed from a comparison between the generated text and ground truth, rather than relying on training probabilities that are not always aligned with the desired metrics. Additionally, we introduce a candidate augmentation to avoid the limitations of a confined candidate set. This broadens the scope of potential dialogues and better captures the richness and diversity of an agent's persona.

## 3 Methodology

In this section, we introduce the proposed LAPDOG model.

### 3.1 Task Formulation

In a persona-based conversation session denoted by $C = \{P, U\}$, the persona $P = \{p_1, \ldots, p_e\}$ consists of $e$ profile sentences providing background information about a machine interlocutor $m$ and the dialogue context $U = \{u_{h,1}, u_{m,1}, ..., u_{h,n}\}$ encompasses the exchange of utterances between a human interlocutor $h$ and the machine interlocutor $m$. In the task of persona-based dialogue generation, the persona $P$ is used to characterize the machine interlocutor $m$, but it contains only four to five sentences, i.e., $4 \leq e \leq 5$. The conversation always starts by the human interlocutor $h$. The primary objective of this task is to generate the response $r = u_{m,n}$ based on the given persona $P$ and dialogue context $U$.

The persona $P$ is short and hence cannot give a full characterization for the background information. To enrich the persona, we utilize a retrieval corpus $\mathcal{D}$ consisting of stories from a story dataset (e.g., ROCStory (Mostafazadeh et al., 2016)). Note that there is no explicit annotation between $\mathcal{D}$ and $P$, necessitating an alternative approach to evaluate the usefulness of the retrieval content.

### 3.2 The Architecture

As shown in Figure 1, the architecture of the LAPDOG model consists of a generator, which adopts a transformer-based encoder-decoder structure, to

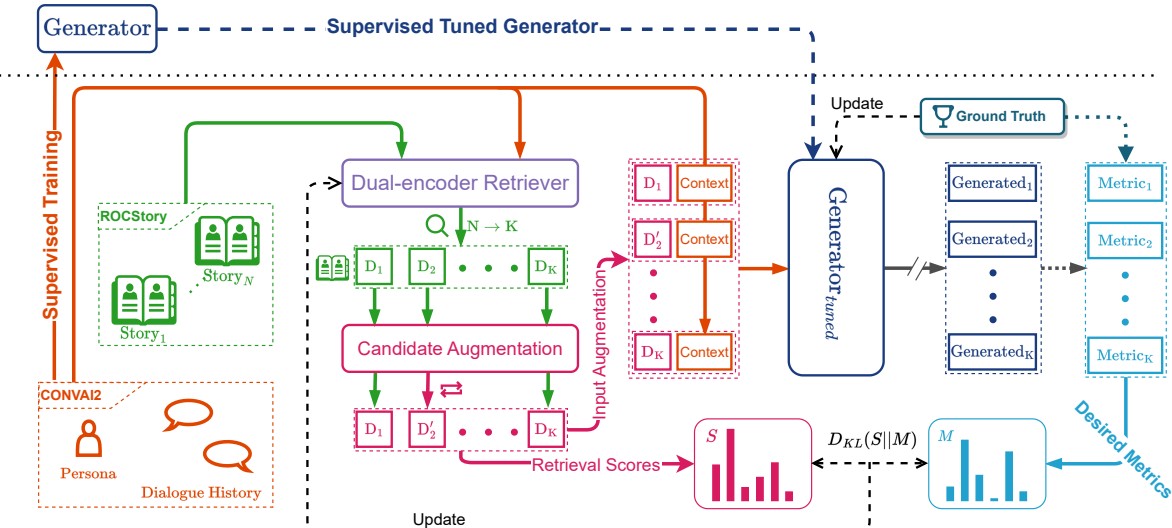

Figure 1: An illustration of the two-stage training process in the proposed LAPDOG model.

generate dialogues and a dual-encoder retriever to efficiently obtain relevant information from an external story corpus.

**Retriever** Based on (Karpukhin et al., 2020), the retriever adopts a transformer-based encoder to embed the query and the story corpus, respectively. The retriever then calculates the dot product similarity score between the query and each story via their average pooled embeddings. Stories with the $K$ highest similarity scores are retrieved.

**Generator** The generator takes a transformer-based encoder-decoder architecture to generate the response from the persona, dialogue history, and retrieved contents. To integrate the retrieved contents with the persona and dialogue history, we leverage the Fusion-in-Decoder (FiD) technique (Izacard and Grave, 2021). Specifically, each retrieved story is combined with the persona and context and individually encoded. The resulting encoded contexts are concatenated and cross-attended in the decoder to generate the final response.

### 3.3 Training Process

It is straightforward to directly train the generator and retriever using the generator's probability distribution in a way similar to the RAG method. However, this strategy does not work well since the retriever would trap into a fixed candidate set and the predicted probability distribution is not always aligned with the desired objectives in the personalized dialogue generation task. Hence, as depicted in Figure 1, the LAPDOG model adopts a two-stage training procedure. In the first stage, the training process starts with supervised training for

the generator (refer to Section 3.3.1). In the second stage, the framework starts to tune the retriever and learn the retrieval augmentation jointly. To learn retrieval augmentation (refer to Section 3.3.2), the retriever's loss is computed from the evaluation metrics between the output of the generator and the ground truth. During the process of learning retrieval augmentation, to prevent the retriever from stagnating around a limited set of candidates, we design the retrieval candidate augmentation (refer to Section 3.3.3), a method ensuring diversity in the retrieval process. Afterward, we enrich the input of the generator with retrieval-enhanced data and compute a generator loss based on the augmented input (refer to Section 3.3.4). Finally, we combine losses from both the generator and retriever to jointly train the two components (refer to Section 3.3.5). In the following sections, we introduce each part in detail.

#### 3.3.1 Supervised Training

First, we train a generator that accepts persona $P$ and context $U$ as input, and the ground-truth response $r$ as the target without involving any retrieval results. Hence, this stage is to minimize the negative log-likelihood, which is formulated as

$$\begin{aligned} \mathcal{L}_{NLL} &= -\log(\mathcal{G}_\theta(r|P, U)) \\ &= -\sum_{i=1}^{|r|} \log(\mathcal{G}_\theta(r_t|P, U, r_{<t})), \end{aligned} \quad (1)$$

where $r_t$ denotes the $t$-th token in $r$, $r_{<t}$ denotes the sequence containing the first to $(t-1)$-th tokens in $r$, $\mathcal{G}_\theta(\cdot)$ denotes the predicted probability distribution of the generator, and $\theta$ denotes parameters of the generator.

After supervised training, we obtain a supervised tuned generator denoted by $\mathcal{G}_{sup}$.

### 3.3.2 Learning Retrieval Augmentation

Intuitively, with the retrieval content as an augmentation, the goal is to improve the generated content in terms of desired metrics. However, it is hard to build direct connections between retrieval contents and the quality of the final generated response to update the retriever. To achieve that, we use the trained generator $\mathcal{G}_{sup}$ as an evaluator to give feedback.

Specifically, given the metric values from the trained generator $\mathcal{G}_{sup}$, we use those metric values as feedback to guide the update of the retriever. In other words, if the generator $\mathcal{G}_{sup}$ finds that the retrieved story $d_i \in D_q$ is useful to improve the performance in terms of the given metrics, we should encourage the retriever to rank the score of $d_i$ to be higher. In this way, we can let the model decide the usefulness of the retrieval content and avoid relying on the retrieval annotations between query $q$ and story $d_i$. However, since the whole generation and metric calculation process is hard or even impossible to be differentiate, we cannot directly perform gradient descent with respect to the calculated metrics to update the retriever. To solve this problem, instead we transform the metric values into a probability distribution as

$$p_i = \frac{\exp\left(\frac{1}{\tau_g}\mathrm{M}(y, \mathrm{Gen}(\mathcal{G}_{sup}, (d_i, P, U)))\right)}{\sum_{c=1}^{K} \exp\left(\frac{1}{\tau_g}\mathrm{M}(y, \mathrm{Gen}(\mathcal{G}_{sup}, (d_c, P, U)))\right)},$$

where $\mathrm{M}(y, \hat{y})$ denotes a metric function to evaluate the quality of the generated text $\hat{y}$ given the ground truth $y$, $\mathrm{Gen}(\mathcal{G}_{sup}, (d_i, P, U))$ denotes the decoded text generated by $\mathcal{G}_{sup}$ given $(d_i, P, U)$ as the input, and $\tau_g$ is a temperature hyperparameter to control the sensitivity of the metric. Here the metric function satisfies that a higher value of $\mathrm{M}(\cdot, \cdot)$ indicates better performance. If a smaller value of $\mathrm{M}(\cdot, \cdot)$ indicates better performance, we can replace $\mathrm{M}(\cdot, \cdot)$ with $-\mathrm{M}(\cdot, \cdot)$ in the calculation of $p_i$. It is easy to see that a useful $d_i$ will have a large $p_i$ and hence $p_i$ can be used as a supervised signal to guide the learning of the retriever. That is, we could make the similarity score returned by the retriever close to $P_{\mathcal{R}} = \{p_i\}_{i=1}^{K}$. Formally, suppose we have top-$K$ retrieval stories $D_q$ with its retrieval scores $S_q \in \mathbb{R}^K$ with respect to the query $q$, we can minimize the KL divergence between $S_q$ and $P_{\mathcal{R}}$ as

$$\mathcal{L}_{\mathcal{R}} = \mathrm{KL}(P_{\mathcal{R}}, \sigma(S_q/\tau_s)), \tag{2}$$

where $\mathrm{KL}(\cdot, \cdot)$ denotes the KL divergence, $\sigma(\cdot)$ denotes the softmax function, and $\tau_s$ is a temperature hyperparameter to control the sensitivity of the similarity scores. Combining $\mathcal{L}_{\mathcal{R}}$ with the retrieval candidate augmentation introduced in the next section, we can update the retriever.

### 3.3.3 Retrieval Candidate Augmentation

During the training process, there is a risk that the retriever gets stuck in a local optimum and consistently retrieves a fixed set or a narrow range of candidates. Consequently, the generator fails to learn from the retriever and disregards the retrieved content. To address this challenge, we design retrieval candidate augmentation to incorporate randomly sampled stories to encourage the framework to explore a wider range of candidates. Specifically, we first replace each $d_i$ with a randomly selected candidate $d_i^{aug}$ at a probability of $\rho$ as

$$d_i^{aug} = \mathrm{CandAug}(d_i, \rho); d_i \in D_q,$$

where $D_q$ denotes the set of retrieval stories, and forms a perturbed set $D_q^{aug} = \{d_i^{aug}\}_{i=1}^{K}$. Then we can compute the dot product similarity between the query $q$ and each $d_i^{aug}$ as the retrieval scores $S_q^{aug} = \{s_{q,i}^{aug}\}_{i=1}^{K}$, where $s_{q,i}^{aug}$ denotes the dot product similarity between $q$ and $d_i^{aug}$. Then we can apply the learning retrieval augmentation to $S_q^{aug}$ and based on Eq. (2) minimize the following loss to update the retriever as

$$\mathcal{L}_{\mathcal{R}}^{aug} = \mathrm{KL}(P_{\mathcal{R}}, \sigma(S_q^{aug}/\tau_s)).$$

### 3.3.4 Training Retrieval-Augmented Generator

With the retrieval content obtained by the retriever, we hope to generate the response more accurately and hence we can further supervised train the generator in a way similar to the first stage (i.e., Section 3.3.1). Specifically, we can minimize the negative log-likelihood of the response given the persona, dialogue context, and retrieval content as

$$\begin{aligned}\mathcal{L}_{\mathcal{G}} &= -\log(\mathcal{G}_\theta(r|P, U, D_p^{aug})) \\ &= -\sum_{i=1}^{|r|}\log(\mathcal{G}_\theta(r_t|P, U, D_P^{aug}, r_{<t})).\end{aligned} \tag{3}$$

It is easy to see that Eq. (3) is similar to Eq. (1) with additionally inputting the retrieval content.

### 3.3.5 Retriever-Generator Joint Training

At the final stage, we aim to jointly train the retriever and generator to further improve them. Specifically, we minimize the sum of the losses of the two components as

$$\mathcal{L} = \mathcal{L}_{\mathcal{R}}^{aug} + \mathcal{L}_{\mathcal{G}}. \tag{4}$$

In Eq. (4), the two loss functions are treated equally. Generally speaking, introducing and tuning a weighting hyperparameter between the two losses may result in better performance but it incurs computational costs when tuning it. For simplicity, we did not introduce this hyperparameter and this could be left for future study.

To summarize, Algorithm 2 in the appendix describes the complete two-stage training process.

### 3.4 Inference Process

During inference, stories from the ROCStory dataset are fetched in alignment with the provided persona and then integrated into the dialogue context using the Fusion-in-Decoder (FiD) technique (Izacard and Grave, 2021). Each combination of story, persona, and context is individually encoded. These encoded contexts are concatenated and processed via cross-attention in the decoder to produce the final response in an auto-regressive fashion. Additional experiments evaluating the effects of various query combinations, such as persona+context and context alone, are detailed in Appendix A.2 to highlight their impact on performance.

## 4 Experiment

In this section, we empirically evaluate the proposed LAPDOG model.

### 4.1 Dataset

ConvAI2 (Dinan et al., 2019) is a dialogue dataset collected from the crowd, featuring 8939/1000 multi-turn conversations that rely on 1155/100 persona descriptions for the train/dev splits. Each persona is succinctly depicted by approximately 5 profile sentences. Paired workers engaged in interactive conversations based on predefined personas.

### 4.2 Retrieval Corpus

Given the absence of a paired annotated retrieval corpus connected to ConvAI2, we employ ROC-Story (Mostafazadeh et al., 2016) as an auxiliary retrieval dataset. Our aim is for the narratives within this dataset to serve as supplemental content to the existing personas within the dialogue. We have undertaken pre-processing of the ROCStory to align the narrative style more closely with persona representation, including changes like transforming 'he' to 'I' and 'does' to 'do'. The detailed pre-processing is listed in Appendix A.7. Statically, there are 98,161 stories within the corpus, and each story is composed of 5 sentences.

### 4.3 Experimental Settings

We employ T5 series models (Raffel et al., 2020) (small, base, XL) as the foundational model used for the generator. We initialize our generator with pre-trained weights from T5 and subsequently fine-tune it on the CONVAI2 dataset as $\mathcal{G}_{sup}$. The dense retriever is initialized with Contriever (Izacard et al., 2021), a dual-encoder retriever that shares a similar encoder structure to BERT (Devlin et al., 2019). We performed fine-tuning on both the retriever and generator using $\mathcal{G}_{sup}$, with a learning rate of $5 \times 10^{-4}$ and $\rho = 0.5$. We further tune for learning retrieval augmentation based on the supervised foundation models in one epoch. We use persona profile as the query to retrieve the relevant stories.

### 4.4 Evaluation Metric

LAPDOG aims to optimize for some generation metrics to enhance the dialogue quality. The evaluation comprises three metrics. The first metric is F1, which computes the harmonic mean of precision and recall on a word level between the generated text and the ground truth. The second metric is BLEU (Papineni et al., 2002; Post, 2018), an $n$-gram precision-based measure that quantifies the overlap between the generated text and the ground truth by penalizing for overly long or short outputs. The third metric is ROUGE-L (Lin, 2004), a variant of ROUGE that considers the longest common subsequence between the generated text and the ground truth, to effectively measure sentence-level structural similarity. With those metrics, we ensure a comprehensive assessment of the quality of generated dialogues. To enhance the aforementioned three metrics, we sum these three metrics together as the overall metric to train the LAPDOG model.

### 4.5 Baseline

To compare the enhancement between different retrieval-augmentation approaches, we compared the results with the following baselines. First, we compare the LAPGOG with T5$_{sup}^{S/B/XL}$ models,

| Model | F1 | BLEU | ROUGE-L | F1↑ | BLEU↑ | ROUGE-L↑ | AVG↑ |
|---|---|---|---|---|---|---|---|
| $T5^S_{Sup}$ | 13.90 | 3.03 | 14.15 | 0.00% | 0.00% | 0.00% | 0.00% |
| $T5^S_{Sup}+\mathcal{R}_{fix}$ | 14.09 | 3.08 | 14.00 | 1.37% | 1.59% | -1.03% | 0.64% |
| $T5^S_{Sup}+\mathcal{R}_{fix}$+RL | 2.50 | 0.21 | 5.49 | -81.98% | -93.07% | -61.22% | -78.76% |
| $T5^S_{Sup}$+RAG | 14.20 | 2.94 | 14.10 | 2.14% | -2.84% | -0.35% | -0.35% |
| $T5^S_{Sup}$+LAPDOG | **14.62** | **3.23** | **14.44** | **5.17%** | **6.57%** | **2.07%** | **4.60%** |
| $T5^B_{Sup}$ | 15.47 | 3.42 | 14.93 | 0.00% | 0.00% | 0.00% | 0.00% |
| $T5^B_{Sup}+\mathcal{R}_{fix}$ | 14.36 | 3.34 | 15.05 | -7.18% | -2.31% | 0.82% | -2.89% |
| $T5^B_{Sup}$+LAPDOG | **16.08** | **3.53** | **15.33** | **3.94%** | **3.21%** | **2.67%** | **3.27%** |
| $T5^{XL}_{Sup}$ | 16.22 | 3.55 | 15.55 | 0.00% | 0.00% | 0.00% | 0.00% |
| $T5^{XL}_{Sup}+\mathcal{R}_{fix}$ | 16.27 | 3.36 | 15.02 | 0.31% | -5.32% | -3.41% | -2.81% |
| $T5^{XL}_{Sup}$+LAPDOG | **17.11** | **3.56** | **15.64** | **5.49%** | **0.30%** | **0.56%** | **2.11%** |

Table 1: Experimental results of various methods based on language models with varying sizes. ↑ denotes the relative improvement over the supervised foundation model. The best result under each setting is shown in bold.

where $S/B/XL$ indicates the model size *small, base, XL* respectively. $T5^{S/B/XL}_{sup}$ serves as the foundation models $\mathcal{G}_{sup}$. We also add a fixed retriever $\mathcal{R}_{fix}$ initialized from Contriever to validate the effectiveness of tuned and untuned retrievers. Meanwhile, we utilized the reinforcement learning tuning ($T5^S_{Sup}+\mathcal{R}_{fix}$+RL) as one baseline, where the reward is set as the desired objective. Lastly, we introduce the RAG tuning that updates the retriever based on the generator's training output probabilities instead of the desired metric, which is to validate the direct and indirect metric tuning for the objective.

## 4.6 Results

The $T5^S_{Sup}$ model forms our baseline. Augmenting it with a fixed retriever, $T5^S_{Sup}+\mathcal{R}_{fix}$, shows a slight improvement in F1 and BLEU scores but a small decrease in the ROUGE-L score. This indicates a moderate enhancement in both F1 and BLEU, though the decrease in ROUGE-L suggests a trade-off in terms of capturing long-distance dependencies.

The results for reinforcement learning tuning, $T5^S_{Sup}+\mathcal{R}_{fix}$+RL, exhibit a significant degradation across all metrics, indicating that this method might be not so effective for this task. This could be due to the challenge of setting an appropriate reward function for reinforcement learning.

The $T5^S_{Sup}$+RAG model slightly outperforms the baseline model in terms of F1 but performs worse in terms of BLEU and ROUGE-L. This suggests that while the model seems to generate more correct words, there may be a compromise on the overall grammatical and semantic quality

| Method | BLEU | ROUGE-L | F1 |
|---|---|---|---|
| LAPDOG | 3.23 | 14.44 | 14.62 |
| w/o BLEU | 3.07 | 14.35 | 14.29 |
| w/o F1 | 2.87 | 13.88 | 13.88 |
| w/o ROUGE-L | 2.96 | 14.12 | 13.99 |

Table 2: Ablation study with respect to the use of metrics in the LAPDOG model.

of the generated text. In contrast, the LAPDOG-enhanced model, $T5^S_{Sup}$+LAPDOG, shows the highest improvements in all metrics among small models. This indicates that LAPDOG significantly enhances the ability to generate high-quality text and captures the desired metrics more effectively than other models.

For larger models, similar phenomena are observed. LAPDOG consistently delivers the best improvements over the base model, no matter whether it is $T5^B_{Sup}$ or $T5^{XL}_{Sup}$. This suggests that the efficacy of LAPDOG is not confined to smaller models and scales well with the model size.

## 5 Ablation Studies

We conduct ablation studies with respect to the metrics, the number of candidates, candidate augmentation, and training strategy, respectively.

## 5.1 Analysis on Metrics

To understand the individual contribution of each metric, we perform ablation experiments by successively removing one metric from the combined optimization process with results shown in Table 2. When we remove BLEU, the performance experiences a slight drop across all metrics with BLEU

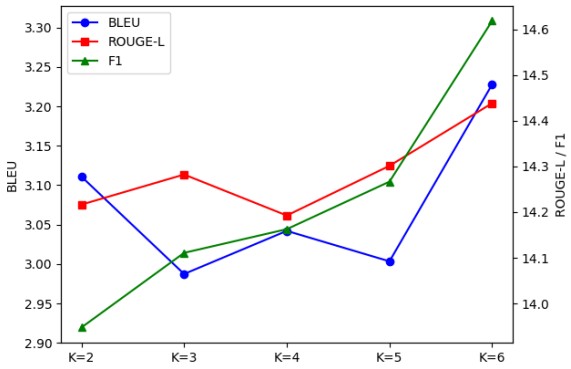

Figure 2: Ablation study with respect to the number of candidates $K$.

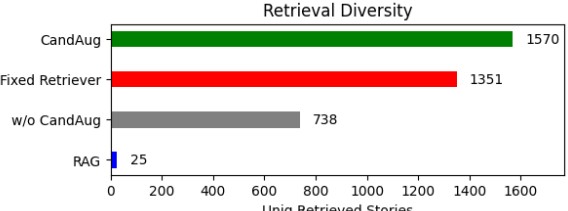

Figure 3: Comparison of the unique number of stories retrieved by different models.

| Method | BLEU | ROUGE-L | F1 |
|---|---|---|---|
| LAPDOG | 3.23 | 14.44 | 14.62 |
| w/o CandAug | 3.14 | 14.19 | 14.43 |
| LAPDOG$_{scratch}$ | 2.95 | 13.90 | 14.23 |

Table 3: Ablation study on the candidate augmentation and training strategy.

decreasing to 3.07, ROUGE-L to 14.35, and F1 to 14.29, suggesting that BLEU contributes to a more precise matching of response generation. Moreover, when we exclude F1, we see a more significant reduction in the performance, indicating the crucial role of F1 to ensure the overlap of words between the generated responses and the ground truth. Lastly, the removal of ROUGE-L also results in a decrease in the performance across all three metrics, showing its essential role in evaluating the coherence of generated dialogues. In summary, each metric contributes uniquely to the optimization process, and their combination in LAPDOG provides a more comprehensive guide for the generation of high-quality, personalized dialogues.

## 5.2 Number of the Candidates

As shown in Figure 2, where $K$ increases from 2 to 6, we observe a consistent improvement in all metrics (i.e., BLEU, ROUGE-L, and F1). Generally, increasing the number of retrieval candidates improves the performance of the model, as evidenced by the improvements in the BLEU, ROUGE-L, and F1 scores. Interestingly, it is observed that the model performance does not monotonically increase with the number of candidates. The performance fluctuates as $K$ varies, implying that the number of retrieval candidates needs to be carefully selected. Too few candidates may not provide enough information for generating responses, while too many ones may introduce irrelevant information, which could potentially confuse the model.

## 5.3 Candidate Augmentation

The influence of candidate augmentation is explored in two aspects: quantitative performance and the diversity of retrieved stories. Table 3 provides a comparison between the performance of the LAPDOG model with and without candidate aug-

mentation. The incorporation of candidate augmentation leads to superior performance across all three evaluation metrics. Specifically, LAPDOG without candidate augmentation attains slightly lower scores in all three metrics. This indicates that the inclusion of candidate augmentation enhances the overall performance of our model, confirming its crucial role in the proposed LAPDOG model.

To further investigate the impact of candidate augmentation to the retrieval diversity, we count the number of unique stories retrieved during testing. As shown in Figure 3, the LAPDOG method with candidate augmentation retrieves 1570 unique stories, whereas without it, the model only retrieves 738 unique stories. This result implies that candidate augmentation significantly contributes to the retrieval diversity. Other methods like RAG and fixed retriever manage to retrieve 25 and 1351 unique stories, respectively. This underlines the effectiveness of the proposed candidate augmentation approach in enhancing the diversity of the retrieval process, which in turn can help generate more personalized and contextually rich responses.

## 5.4 Training Strategy

In contrast to the two-stage training process, we perform an ablation study by training LAPDOG from scratch, bypassing the first stage. As shown in Table 3, the results exhibit a significant decrease in all the metrics compared with the two-stage approach. Additionally, training directly from scratch requires more time to converge when compared with the two-stage training process. Overall, the two-stage training process is essential from both performance and efficiency standpoints.

# 6 Conclusion

In this paper, we introduced LAPDOG, an end-to-end learnable retrieval augmentation personalized dialogue generation framework. We show that LAPDOG jointly tunes the retriever with the generator to retrieve useful stories from the ROCStory dataset for enhancing the desired performance over the CONVAI2 dataset. LAPDOG gains consistent performance enhancement over language models with varying sizes.

## Acknowledgements

This work is supported by NSFC general grant 62076118 and Shenzhen fundamental research program JCYJ20210324105000003.

## Limitations

Given resource constraints, in this paper, we employ language models such as T5 and do not conduct experiments based on currently prevalent large language models (Brown et al., 2020; Ouyang et al., 2022). Recognizing the enhanced reasoning capabilities of large language models, we posit that tuning the retrieval content with such models could yield significant advantages. Additionally, due to resource limitations, we study a small number of extracted passages (i.e., 2-6) and a short context length (i.e., 512 tokens). Nevertheless, we anticipate that incorporating a larger set of integrated stories and a longer context would further enhance the performance. Also, a more diverse objective rather than the summation of F1, ROUGE, and BLEU might be more helpful to train an engaging conversational AI system. Also, the generator is simply a conventional T5 model rather than explicitly designed models, which could help improve the performance of the proposed LAPDOG model further.

## Ethics Statement

This work proposes a novel LAPDOG model for personalized dialogue generation, focusing on generating highly tailored responses by leveraging persona profiles and dialogue context. As with all machine learning applications, it is crucial to consider the ethical implications. The use of personal information in our model is limited to fictional persona profiles, and we do not handle or store any real personal data in our experiments. However, when applying our model to real-world applications, careful consideration should be given to data privacy and consent. It is essential to ensure that all personal information used to generate personalized dialogues is obtained ethically and used with the individuals' informed consent. Moreover, the generated content should respect user privacy, dignity, and cultural sensitivities.

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

# A Appendix

## A.1 Detail Settings for Training

| Setting | Value |
|---------|-------|
| $\tau_g$ | 0.85 |
| $\tau_s$ | 0.8 |
| $dialog_{maxturns}$ | 3 |
| Dropout | 0.1 |
| LR | 5.00E-04 |
| Optimizer | Adam |
| WeightDecay | 0.01 |
| RetrievalCandidate | 6 |
| $\rho$ | 0.5 |
| BatchSize | 240 |

Table 4: Detail experimental settings for training LAP-DOG.

## A.2 Query Analysis

As shown in Table 5, we aim to analyze the performance impact of different retriever queries. As indicated in Table 5, using the *Persona* alone as a query achieves the highest BLEU score, albeit with a slight trade-off in ROUGE-L and F1 scores. When combining *Persona* with the *Dialogue*, the ROUGE-L and F1 scores improve marginally, but at the expense of a slightly decreasing in BLEU. Following the idea of forward retrieval (Jiang et al., 2023), we experimented with using the generator's output (*Generated*) as a query, but observed less competitive performance. Lastly, a strategy of using a single sentence from the persona profile chosen at random (*One Persona*) resulted in worse performance on all metrics.

## A.3 Evaluation Results on CONVIAI2 Revised Dataset

As shown in Table 6, LAPDOG consistently enhances the performance on the revised dataset, where the persona is paraphrased to more implicit background sentences. The revised version is considered a more difficult task than the original task.

## A.4 Case Study

As shown in Tables 11 and 12, we present the case studies on two conversations to compare the generation results among the non-retrieval-augmented results of $T5_{sup}^{XL}$, retrieval augmented results of $T5_{sup}^{XL}$+LAPDOG, and the ground truth.

For the conversation in Table 11, the agent is going to talk about divorce as indicated in the ground truth. LAPDOG retrieves several stories about the bad days with his life and family, and this could be a clue for him to decide to divorce. In this conversation, $T5_{sup}^{XL}$ complains about his life but does not mention anything about divorce. LAPDOG, reinforced by the retrieval stories and persona, has a stronger intention to generate the divorce decision, which would be more aligned with the intention in the ground truth. The other retrieved stories describe the messes that happened during the working time, which may reflect the persona *"I hate my job."*

In the conversation mentioned in Table 12, the conversation is a simple start with a *"How are you doing today?"*, $T5_{sup}^{XL}$ gives a standard, safe, but bland answer as *"I am good. How are you?"*, while LAPDOG incorporates stories and persona about the gym to answer with *"I'm good. Just got back from the gym. How are you?"*, which would be more information-intensive and engaging. Additionally, the story *"Lifestyle Change"* provides a good clue for the agent about why he decided to go to the gym, and the story *"Home Gym"* describes the enthusiasm about the workout. These would potentially provide the model with enriched information on generating personalized responses.

As shown in Table 13, the $T5_{sup}^{XL}$ model simply replied with a *"I hope so!"* without further informative content, while LAPDOG answers with richer information as *"I hope so! I'm only in grade 3 so I'm hoping to go to Disney World soon!"*, which might consider the information from both persona and stories. Additionally, the retrieved stories like *"Dream Job"* or *"Disneyland"* are aligned with the agent's favor.

| Query | BLEU | ROUGE-L | F1 |
|-------|------|---------|-----|
| Persona | 3.23 | 14.44 | 14.62 |
| Persona+Dialogue | 3.19 | 14.83 | 15.06 |
| Generated | 3.05 | 14.22 | 14.32 |
| One Persona | 3.10 | 14.29 | 14.39 |

Table 5: The evaluation on different queries.

| Method | BLEU | ROUGE-L | F1 |
|--------|------|---------|-----|
| $T5_{Sup}^{S}$ | 2.01 | 12.48 | 11.56 |
| $T5_{Sup}^{S}$+LAPDOG | 2.21 | 13.44 | 12.82 |

Table 6: Evaluation results on CONVIAI2-Revised version, where the persona is paraphrased to more implicit background sentences. The revised version is considered as a more difficult task than the original task.

| Story Examples |
| --- |
| **Antique Car Show** |
| I like fixing cars. |
| I have just finished repairing and restoring an antique sports car. |
| I proudly enter it in a local car show for antique vehicles. |
| I won a cash prize for my hard work! Now I have enough money to buy another antique car to restore. |
| **Mechanic** |
| I am a mechanic and love to work on cars. |
| I work in a shop three days a week. |
| In my spare time I fix cars for people in my garage. |
| I do great work at a fast pace for a small fee. |
| I get two incomes doing what I love. |

Table 7: Two examples of stories, pre-processed from the ROCStory dataset.

---

**Algorithm 1** Pre-processing Procedure of Story Corpus

---

**Input:** A story corpus $\mathcal{D}_{ori}$
**Output:** A pre-processed story corpus $\mathcal{D}$
1: Extract named entity using *BERT-BASE-NER* from $\mathcal{D}_{ori}$
2: Filter out person-related named entity with tag *B-PER*
3: Replace the person-related named entity with the first-person narratives within stories in $\mathcal{D}_{ori}$
4: Process the first-person stories over a grammatical error correct model *gec-t5_small*
5: Output the corrected stories as $\mathcal{D}$

---

**Algorithm 2** Complete Training Process of LAPDOG

---

**Input:** Persona sentences $P$, dialogue context $U$, a ground truth response $y$, a generator $\mathcal{G}$, a dense retriever $\mathcal{R}$, and a story corpus $\mathcal{D}$
**Output:** A tuned retriever $\mathcal{R}_{tuned}$, a tuned generator $\mathcal{G}_{tuned}$
1: Construct query $q$ from a query function $Query(P, U)$
2: **Stage1:**
3: Initialize $\mathcal{G}_{sup}$ with $\mathcal{G}$
4: Train a supervised tuned generator $\mathcal{G}_{sup}$ given input $(P, U)$ and ground truth $y$
5: Train $\mathcal{G}_{sup}$ until converge
6: **Stage2:**
7: Retrieve top-$K$ stories $D_q$ given $q$
8: Apply Candidate Augmentation $D_q^{aug} = \text{CandAug}(d_i, \rho); d_i \in D_q$
9: Compute retriever scores $S_q^{aug}$ between query $q$ and $D_q^{aug}$
10: **for** Retrieved story $d_i$ in $D_q^{aug}$ **do**
11:     Construct augmented input by concatenation $a_i = [d_i; P; U]$
12:     Generate text by $\text{pred}_i = \mathcal{G}_{sup}(a_i)$
13:     Compute metrics as $m_i = \text{M}(\text{pred}_i, y)$
14: **end for**
15: Gather $M = \{\text{softmax}(m_i)\}_{i=1}^{K}$, $A = \{\text{softmax}(a_i)\}_{i=1}^{K}$
16: Compute retriever's loss by $\mathcal{L}_{\mathcal{R}}^{aug} = \text{KL}(M, S_q^{aug})$
17: **for** augmented input $a_i$ in $A$ **do**
18:     Compute negative log-likelihood loss $\mathcal{L}_{\mathcal{G}}$ with input $a_i$ and ground truth target $y$
19:     Update $\mathcal{G}_{sup}$ and $\mathcal{R}$ by $\mathcal{L} = \mathcal{L}_{\mathcal{R}}^{aug} + \mathcal{L}_{\mathcal{G}}$ as $\mathcal{G}_{tuned}$ and $\mathcal{R}_{tuned}$
20: **end for**
21: Repeat the steps in Stage2 until converge

---

## A.5 Comparison to Traditional Knowledge Dialogue Approaches

Evaluating LAPDOG against traditional knowledge-grounded dialogue methods is crucial. A comparison with *"Low-Resource Knowledge-Grounded Dialogue Generation"* (Zhao et al., 2020) was considered but not possible due to the unavailability of its code. Instead, we selected ITDD (Incremental Transformer with Deliberation Decoder) (Li et al., 2019) for this purpose, given its proven effectiveness and wide recognition.

### A.5.1 ITDD Experimental Setup

LAPDOG and ITDD are based on different principles. LAPDOG uses an unsupervised approach to train both a retriever and a generator for extracting relevant content from an external corpus. On the other hand, ITDD is designed to merge pre-annotated document-conversation pairs. To ensure a fair comparison, we used an off-the-shelf retriever (Izacard et al., 2021) to create paired data from the persona and ROC story corpus for the ITDD model.

### A.5.2 Comparison Results

We compared LAPDOG and ITDD using various metrics, and the results are presented in the table below, showcasing LAPDOG's superior performance.

| Method | F1 | BLEU | ROUGE-L |
|---|---|---|---|
| ITDD | 9.71 | 0.66 | 10.90 |
| T5-S+LAPDOG | 14.62 | 3.23 | 14.44 |
| T5-B+LAPDOG | 16.08 | 3.53 | 15.33 |
| T5-L+LAPDOG | 17.11 | 3.56 | 15.64 |

Table 8: Comparison of LAPDOG and ITDD on key performance metrics.

These results highlight LAPDOG's effectiveness compared to the traditional ITDD method, enhancing the paper's overall assessment of LAPDOG's performance.

### A.6 Complete Training Procedure

The complete training procedure is described at Algorithm 2.

### A.7 Pre-process ROCStory Coprus

The pre-processing procedure is described at Algorithm 1.

## A.8 Extended Evaluation Metrics

The evaluation metrics for LAPDOG have been broadened to incorporate METEOR and BERT scores. This enhancement supplements the foundational assessment based on F1, BLEU, and ROUGE-L metrics, presenting a more diverse evaluation landscape. The additional evaluation outcomes are tabulated in Table 9.

Based on the results in Table 9, LAPDOG excels in the METEOR score relative to baseline models, showcasing its capability in nuanced linguistic comprehension. However, the variance in BERT scores is minimal, likely due to LAPDOG's optimization for traditional metrics. Enhancing performance by tailoring optimization for BERT scores represents a promising area for future inquiry.

### A.9 Additional Related Work on Large Language Models (LLMs)

Language models compute probability distributions over text sequences. Recent advancements have escalated these models from millions of parameters (Radford et al., 2019) to billions (Brown et al., 2020), extending the training corpus to encompass web texts and instructional data (Ouyang et al., 2022). These strides have significantly enhanced the performance of large language models (LLMs) across a myriad of NLP tasks. Notably, in conversational tasks, the quality of generated text improves with the expansion of both the model size and training corpus. Our proposed approach, LAPDOG, diverges from the prevailing trend of scaling; it leverages retrieval-augmented generation to yield more diverse and interpretable results, albeit with smaller model parameters and corpus size. Despite employing a smaller model in our experiments, we posit that our adaptive retrieval approach could complement existing LLMs, thereby potentially elevating their result-generation efficacy.

### A.10 Human Evaluation

We conducted a human evaluation to gauge the preference between the retrieval-augmented LAPDOG and the fine-tuned T5 model. Evaluators were presented with a dialogue accompanied by two responses from each model and were asked to choose their preferred response.

Referring to Table 10, while LAPDOG was preferred by 56.32% of the evaluators, the difference between the two models is relatively narrow. This indicates that both models have their merits in

| Model | METEOR | BERT-F1 | BERT-PRECISION | BERT-RECALL |
|---|---|---|---|---|
| T5$_{Sup}^{XL}$ | 16.45 | 85.95 | 87.07 | 84.88 |
| T5$_{Sup}^{XL}$+$\mathcal{R}fix$ | 16.21 | 85.78 | 86.99 | 84.62 |
| T5$_{Sup}^{XL}$+LAPDOG | 17.76 | 85.99 | 87.35 | 84.70 |

Table 9: Additional evaluation metrics, METEOR and BERT scores.

| | LAPDOG | Fine-tuned Model |
|---|---|---|
| Human Preferred | 56.32% | 43.68% |

Table 10: Comparison of human preference for responses generated by LAPDOG and a fine-tuned T5 model.

certain conversational contexts. The slight edge for LAPDOG suggests that retrieval-augmented responses might align closer to human expectations in some scenarios. Delving deeper into the nuances of this preference, especially across varied dialogue topics, would provide a more comprehensive understanding.

| Conversation #1 | |
|---|---|
| Persona | I am 40 years old.
I work as a car salesman.
My wife spends all my money.
I hate my job.
I'm planning on getting a divorce. |
| Dialogue | ...
Agent: I know the struggle, I volunteer at a homeless shelter.
Human: I hope not either, would not wish it on my worse enemy. |
| $T5^{XL}_{sup}$ | I am 40, so I think I am done with it. |
| $T5^{XL}_{sup}$+LAPDOG | Yeah, I feel you. I want to get a divorce. |
| Ground Truth | Between wife, job with no money, I am thinking now of divorce. |
| Retrieval Stories | **Money**
I have a full-time job.
My wife stays at home.
One day, I worked an over-time shift.
I decided to spend all the extra money on myself.
This made my wife sad.
**Randy leaves**
I am a man who has 5 kids.
I started arguing with my wife one day.
I got so upset I promised to leave my wife.
So that's what I did.
I left my children and wife.
**Cruel Boss**
I have been dealing with extra expenses in the family.
Since my wife is a stay-at-home mom, I pay for everything.
I tried to get a raise by talking to my boss the next day.
My boss understood my situation and sincerely apologized.
Although I have been my employee for years, the offer was rejected.
**Paul Gets Fired**
I hate my job.
Every day at work I get angry and say mean things to people.
My boss gave me a verbal warning about my attitude at work.
Currently I am on a performance plan at work.
Next month I will be fired. |

Table 11: The first case study.

| Conversation #2 | |
|---|---|
| Persona | I like to spend my money on cars. |
| | I go to the gym most days just to socialize. |
| | I work from home. |
| | I watch too much tv in spare time. |
| | I have never had a steady relationship. |
| Dialogue | Human: Hi, How are you doing today? |
| $T5^{XL}_{sup}$ | I am good. How are you? |
| $T5^{XL}_{sup}$+LAPDOG | I'm good. Just got back from the gym. How are you? |
| Ground Truth | I'm pretty good. Just a little lonely. |
| Retrieval Stories | **Lifestyle Change** |
| | I was overweight and always tired of not having a girlfriend. |
| | I decided to make some lifestyle changes. |
| | I cancelled my cable TV and joined a gym. |
| | I exercised and ate healthily for five months. |
| | I started to feel good about myself and started to date! |
| | **Home Gym** |
| | I am addicted to working out and I go every day. |
| | I just started a new job that has me working long hours. |
| | These long hours prevent me from being able to go to the gym. |
| | By the time I get off work the gym is closed. |
| | I went out and bought all the stuff I needed to work out at home. |
| | **Mechanic** |
| | I am a mechanic and love to work on cars. |
| | I work in a shop three days a week. |
| | In my spare time I fix cars for people in my garage. |
| | I do great work at a fast pace for a small fee. |
| | I get two incomes doing what I love. |
| | **Car Mechanic** |
| | I love to work on cars. |
| | I work as a mechanic at a local car repair shop. |
| | I decided to work on cars in my spare time too. |
| | I advertised on social media that I would fix cars for cheap. |
| | Before long I had a nice little side business going. |
| | **The Drive-Through** |
| | I ate out every day of my adult life. |
| | I was a 27-year-old man. |
| | I loved going to drive troughs and preferred eating in my car. |
| | One day while I was ordering my favorite food I had a heart attack. |
| | I was rushed to the hospital by the employees and went into a coma. |
| | I died due to my excessive and unhealthy lifestyle later that night. |

Table 12: The second case study.

| Conversation #3 | |
|---|---|
| Persona | I love to sing songs from the movie Frozen. |
| | I am in the third grade. |
| | I play with my friends on the playground. |
| | I love to go to Disney world every year. |
| | Mickey Mouse is my favorite character. |
| Dialogue | ... |
| | Human: They're about 5 years old. |
| | Agent: Yeah Disney is their favorite! We always go to Disney land! |
| | Human: That's great, maybe we will meet there one day! |
| $T5_{sup}^{XL}$ | I hope so! |
| $T5_{sup}^{XL}$+LAPDOG | I hope so! I'm only in grade 3 so I'm hoping to go to Disney World soon! |
| Ground Truth | Mickey Mouse is my favorite, and I love songs from the movie Frozen. |
| Retrieval Stories | **Xena The Shy** |
| | My friend is very shy. |
| | I am always being teased because of my name. |
| | The boys at school call my warrior princess. |
| | People are so mean. |
| | I wish I was a warrior princess so I could make them be quiet! |
| | **Meeting Mickey** |
| | I went to Disney World for the first time when I was 7. |
| | My parents bought me a book to collect character signatures. |
| | I was so excited to get my signature so I didn't notice a line. |
| | I ran to hug I cut in front of many other kids. |
| | The girl who was next in line yelled at me. |
| | **dream job** |
| | I have always wanted to play a character at Disney World. |
| | I moved to Orlando and applied for a job. |
| | Disney hired me as a customer service rep. |
| | I worked very hard to achieve my goal. |
| | The other day I got a promotion to play Mickey Mouse. |
| | **Pageant** |
| | Little I was a three-year-old beauty queen. |
| | I was very good at walking and smiling for the judges. |
| | This week it was different. |
| | I would have to sing. |
| | The day of the show I looked beautiful. |
| | Too bad butterflies in my tummy flew away with the words to the song. |
| | **Cade's Christmas Show** |
| | I am excited about my Christmas show. |
| | I have been practicing Jingle Bells all week. |
| | On the day of the show I was so nervous. |
| | On stage I look out at the audience to find my mom. |
| | I am in the first row, so now I know everything will be fine. |
| | **Disneyland** |
| | I have never been to Disneyland. |
| | I love all the Disney characters. |
| | My mom decided to take me to Disneyland. |
| | I was so excited when I got there. |
| | It was the best day of my young life. |

Table 13: The third case study.