# OpenReview forum: "Learning Retrieval Augmentation for Personalized Dialogue Generation"
_EMNLP/2023/Conference — EMNLP 2023 Main_

### Official Review · Reviewer_Afr1 · 2023-08-05

**Soundness:** 3

**Excitement:**

4: Strong: This paper deepens the understanding of some phenomenon or lowers the barriers to an existing research direction.

**Paper Topic And Main Contributions:**

This paper focuses on personalized dialogue generation. To address the issue of having too few samples in persona profiles, the authors propose LAPDOG, which leverages external story documents for persona dialogue generation. In the proposed method, the retriever and generator are jointly trained, eliminating the need for the proposed method to rely on annotated retrieval datasets. The authors also design a retrieval candidate augmentation during training to ensure diversity in the retrieval process.

**Reasons To Accept:**

1. The authors’ analysis of the difficulties in the personalized dialogue generation task is reasonable. The proposed method effectively addresses these issues through its well-designed approach.
2. The paper is well-written, with a clear structure and easy-to-follow presentation of its contributions.
3. The authors have committed to open-sourcing the code and pre-trained model.

**Reasons To Reject:**

The baselines used in the experiments are not competitive enough. There is a lack of methods proposed by recent cutting-edge research, which makes the comparative experiments less convincing.

**Reproducibility:**

4: Could mostly reproduce the results, but there may be some variation because of sample variance or minor variations in their interpretation of the protocol or method.

**Reviewer Confidence:**

3: Pretty sure, but there's a chance I missed something. Although I have a good feel for this area in general, I did not carefully check the paper's details, e.g., the math, experimental design, or novelty.

---

> ### Author Rebuttal · Authors · 2023-08-29
>
> We greatly thank you for your constructive comments. We address your concerns as follows.
>
> **Question: The baselines used in the experiments are not competitive enough. There is a lack of methods proposed by recent cutting-edge research, which makes the comparative experiments less convincing.**
>
> We acknowledge the reviewer's point about the absence of cutting-edge baselines in our experiments. The unavailability of public code or too complex training stages for some of these cutting-edge approaches influenced our decision to use T5 as our generator architecture for a fair comparison. Our focus is to test the improvements coming specifically from our joint retriever-generator approach, embodied in LAPDOG, rather than the generator architecture itself.
>
> It's important to note that LAPDOG's primary contribution lies in its ability to effectively fuse external story documents into persona dialogues. The framework is modular and not tied to a specific generator architecture, allowing it to be employed in conjunction with various state-of-the-art dialogue models.
>
> Unlike specialized architectures such as Persona-Adaptive Attention (PAA) [1], Coarse-to-Fine Persona-Aware Training Framework [2], and Bert-over-bert [3], LAPDOG offers a “plugin-like” functionality. This means it can be layered on top of these advanced methods, likely boosting their performance by providing additional contextual stories for persona profiles.
>
> In future work, we plan to investigate how LAPDOG can synergize with these advanced architectures to substantiate its utility in enhancing the performance of personalized dialogue generation.
>
> **Reference**
>
> [1] Huang, Qiushi, Yu Zhang, Tom Ko, Xubo Liu, Bo Wu, Wenwu Wang, and H Tang. 2023. “Personalized Dialogue Generation With Persona-Adaptive Attention”. Proceedings of the AAAI Conference on Artificial Intelligence 37 (11):12916-23. https://doi.org/10.1609/aaai.v37i11.26518.
>
> [2] Li, Yunpeng, Yue Hu, Yajing Sun, Luxi Xing, Ping Guo, Yuqiang Xie, and Wei Peng. 2023. “Learning to Know Myself: A Coarse-to-Fine Persona-Aware Training Framework for Personalized Dialogue Generation”. Proceedings of the AAAI Conference on Artificial Intelligence 37 (11):13157-65. https://doi.org/10.1609/aaai.v37i11.26545.
>
> [3] Song, Haoyu, Yan Wang, Kaiyan Zhang, Wei-Nan Zhang, and Ting Liu. "BoB: BERT over BERT for training persona-based dialogue models from limited personalized data." arXiv preprint arXiv:2106.06169 (2021).

---

### Official Review · Reviewer_d41o · 2023-08-05

**Soundness:** 4

**Excitement:**

4: Strong: This paper deepens the understanding of some phenomenon or lowers the barriers to an existing research direction.

**Paper Topic And Main Contributions:**

The paper introduces a model that utilizes context-based stories from an external story corpus to enhance the persona-specific dialogue system. The proposed algorithm comprises two sequential models. The initial model employs a vanilla generative approach with a transformer-based encoder-decoder using inputs such as the persona and user utterance. Subsequently, the second model adopts a retrieval-based generative architecture. It leverages the text generated by the first model to extract the top K retrieval stories. These stories, along with the persona and dialogue history, contribute to the generation of persona-specified text.

**Questions For The Authors:**

> Is the amount of stories you are taking in your dataset enough to cover every type of persona-specified dialogue?

> Why don't you provide other evaluation metrics, BERT score, METEOR, and Human evaluation?

**Reasons To Accept:**

1. The work proposes a new differentiable probability distribution technique to remove the problem of using non-differentiable metric values (to train which story to retrieve for maximum information).

2. A novel approach to generate persona-specific dialogues for a conversation.

3. It solved the problem of getting stuck in local optima in a narrow range of candidates by candidate augmentation.

**Reasons To Reject:**

No such potential reason to reject the paper.

> The model architecture needs to be discussed in detail, particularly the ROC story selection part.

> The related work section should include a discussion of the work in the context of LLMs.

**Reproducibility:**

3: Could reproduce the results with some difficulty. The settings of parameters are underspecified or subjectively determined; the training/evaluation data are not widely available.

**Reviewer Confidence:**

4: Quite sure. I tried to check the important points carefully. It's unlikely, though conceivable, that I missed something that should affect my ratings.

**Typos Grammar Style And Presentation Improvements:**

The paper contains punctual marks and formatting problems, including instances of bold font. Please conduct a thorough review to address these issues.

---

> ### Author Rebuttal · Authors · 2023-08-29
>
> We greatly thank you for your constructive comments. We address your concerns as follows.
>
> **Q1: The model architecture needs to be discussed in detail, particularly the ROC story selection part.**
>
> Thanks for your suggestion. In the revision, we will give more details on model architecture. We did not explicitly do selection on the ROCStory dataset, and we just let the retriever adaptively choose appropriate stories. We will introduce more details in the revision.
>
> **Q2: The related work section should include a discussion of the work in the context of LLMs.**
>
> We appreciate the reviewer's suggestion to discuss our work in the context of Large Language Models (LLMs). Indeed, the rise of LLMs like ChatGPT and LLama1/2 has significantly impacted various tasks in natural language processing, including dialogue generation. We will add a subsection in the “Related Work” section to discuss how our approach both complements and diverges from techniques that utilize LLMs. Specifically, we will address how our model's capability to adaptively retrieve and incorporate external story-based context offers a unique value proposition compared to the more generalized capabilities of LLMs.
>
> **Q3: Is the amount of stories you are taking in your dataset enough to cover every type of persona-specified dialogue?**
>
> We recognize the inherent limitations of any dataset in fully capturing the diversity of human personas and dialogues. Actually, we cannot guarantee that the story dataset can cover every type of persona-specified dialogue. However, our primary focus in this work is to introduce and validate a new methodological framework for enhancing persona-based dialogue systems by leveraging external story datasets. In light of your suggestion, we are interested in constructing large datasets to cover various types of persona-specific dialogues.
>
> **Q4: Why don't you provide other evaluation metrics, BERT score, METEOR, and Human evaluation?**
>
> We appreciate the reviewer's suggestion to consider a more comprehensive suite of evaluation metrics, including BERT score, METEOR, and Human evaluation. Our framework, LAPDOG, is primarily designed to optimize a compound metric including F1, BLEU, and ROUGE-L, and naturally our experiments focus on evaluating models in terms of those metrics.
>
> In light of your suggestion, we have extended our evaluation metrics to include METEOR and BERT scores. The following table summarizes the results.
>
> | Model | METEOR | BERT-SCORE-F1 | BERT-SCORE-PRECISION | BERT-SCORE-RECALL |
> | --- | --- | --- | --- | --- |
> | T5$^{XL}_{Sup}$ | 16.45 | 85.95 | 87.07 | 84.88 |
> | T5$^{XL}{Sup}$+$\mathcal{R}{fix}$ | 16.21 | 85.78 | 86.99 | 84.62 |
> | T5$^{XL}_{Sup}$+LAPDOG | 17.76 | 85.99 | 87.35 | 84.70 |
>
> According to the results, LAPDOG demonstrates superior performance in terms of the METEOR score. This suggests that our framework may better capture some linguistic nuances, such as synonymy and word order, when compared with baseline models, since we optimized for F1, BLEU, and ROUGE-L. However, the BERT scores showed marginal improvement or even degradation. We attribute this to LAPDOG being optimized for traditional statistical metrics rather than neural model-based metrics. We believe that directly optimizing BERT scores in our framework could result in better performance, which could be a future direction of our work.
>
> Due to time constraints during the response period, human evaluations could not be conducted. We acknowledge the importance of such evaluations for a more rounded understanding of our model's capabilities and limitations. These will be included in the revision.
>
> **Q5: The paper contains punctual marks and formatting problems, including instances of bold font. Please conduct a thorough review to address these issues.**
>
> We recognize the importance of clear presentation and have meticulously revised the manuscript to fix typos and formatting issues. Thank you for bringing this to our attention and we will polish the manuscript in the revision.

---

### Official Review · Reviewer_jXtS · 2023-08-06

**Soundness:** 4

**Excitement:**

3: Ambivalent: It has merits (e.g., it reports state-of-the-art results, the idea is nice), but there are key weaknesses (e.g., it describes incremental work), and it can significantly benefit from another round of revision. However, I won't object to accepting it if my co-reviewers champion it.

**Paper Topic And Main Contributions:**

The paper introduces a method called Learning Retrieval Augmentation for Personalized Dialogue Generation (LAPDOG) to tackle the challenge of generating genuinely personalized dialogues with constrained persona profiles. LAPDOG utilizes external knowledge from story data to enhance the persona profiles and enhance the generation of personalized responses. The LAPDOG model consists of two stages: retrieval-augmented generation and joint training of the retriever and generator. In the retrieval-augmented generation stage, a dense retriever is trained to retrieve relevant stories based on a query. The generator is subsequently trained using the retrieved stories as additional input. In the joint training stage, both the retriever and generator are trained together to enhance performance. The training process involves minimizing loss functions and optimizing metrics such as F1, BLEU, and ROUGE-L.

**Questions For The Authors:**

The experiments conducted in the study are not solid enough, as most of the baseline models listed in Table 1 demonstrate negative performance. Furthermore, the paper neglects to compare the proposed approach with some traditional methods of knowledge dialogue that could be readily applied to the current scenario.

**Reasons To Accept:**

The motivation behind this study is to investigate the possibilities of utilizing external textual resources, such as story data, to enhance the information within persona profiles and improve the generation of engaging and contextually relevant dialogues. The paper tackles the challenges associated with explicit annotations for retrieval and the selection of pertinent content for persona profile augmentation. The motivation is reasonable.

**Reasons To Reject:**

The experiments conducted in the study are not solid enough, as most of the baseline models listed in Table 1 demonstrate negative performance. Furthermore, the paper neglects to compare the proposed approach with some traditional methods of knowledge dialogue [1] that could be readily applied to the current scenario.

[1]. Low-Resource Knowledge-Grounded Dialogue Generation

**Reproducibility:**

3: Could reproduce the results with some difficulty. The settings of parameters are underspecified or subjectively determined; the training/evaluation data are not widely available.

**Reviewer Confidence:**

3: Pretty sure, but there's a chance I missed something. Although I have a good feel for this area in general, I did not carefully check the paper's details, e.g., the math, experimental design, or novelty.

---

> ### Author Rebuttal · Authors · 2023-08-29
>
> We greatly thank you for your constructive comments. We address your concerns as follows.
>
> **********Q1: The experiments conducted in the study are not solid enough, as most of the baseline models listed in Table 1 demonstrate negative performance.**********
>
> We acknowledge the reviewer's query about the baseline models' negative performance in Table 1. It's worth noting that personalized dialogue generation is not a straightforward task that can be easily tackled by directly training models on the available data. The need for nuanced contextual understanding, effective persona management, and the seamless integration of external knowledge makes it a challenging problem in the NLP landscape. Hence, a negative performance for baseline models is indicative of these challenges. The negative performance underscores the task's complexity rather than the study's limitations.
>
> The chosen baselines were to isolate the impact of retrieval augmentation, which is a focal point of our study. Each baseline addresses a unique research question:
>
> 1. A generator-only model gauges the retriever's impact.
> 2. An untuned retriever assesses the value of joint tuning.
> 3. Reinforcement learning evaluates the need for metric-based tuning.
> 4. RAG serves as a comparative retrieval-augmented framework, a popular framework often used in retrieval-augmented text generation.
>
> The study's experimental design aims to provide a nuanced understanding of retrieval augmentation. We use T5 as the generator for simplicity, allowing us to focus on the retrieval aspects. Our work adds a unique dimension to the existing solutions by emphasizing learning-based retrieval augmentation. We believe that our LAPDOG framework is promising for augmenting existing SOTA techniques (which usually focus on improving generator solely), a future research direction we are eager to explore.
>
> **Q2: Furthermore, the paper neglects to compare the proposed approach with some traditional methods of knowledge dialogue that could be readily applied to the current scenario.**
>
> We recognize the reviewer's concerns regarding the need for comparison with traditional methods of knowledge-grounded dialogue generation. We aimed to compare LAPDOG with traditional methods, specifically “Low-Resource Knowledge-Grounded Dialogue Generation.” Unfortunately, the absence of publicly available code for this work posed a challenge. Instead, we compared with ITDD (Incremental Transformer with Deliberation Decoder) [1], the most effective baseline method in the above-mentioned paper and a widely used traditional approach, as an additional baseline method. This allows us to evaluate LAPDOG against traditional frameworks, enhancing the rigour of our experimental design and results. The detailed experimental setting and results are introduced as follows and we will add those results in the revision.
>
> - *ITDD Experimental Setup*
>     - Our LAPDOG model and traditional knowledge-grounded dialogue methods operate under different paradigms. Specifically, LAPDOG is designed to unsupervisedly train a retriever alongside a generator to adaptively extract valuable content from an external corpus, while ITDD learns to fuse the document where document-conversation pairs have already been annotated.
>     - To bridge this methodological difference and ensure a fair comparison, we employed an off-the-shelf retriever [2] to create paired data from the persona and the ROC story corpus, which was subsequently fed into the ITDD model.
> - *Comparison Results*
>     - We performed a meticulous comparison between LAPDOG and ITDD, examining their performance in terms of several key metrics. The results, which are shown in the following table, affirm the effectiveness of LAPDOG over ITDD.
>
>     | Method | F1 | BLEU | ROUGE-L |
>     | --- | --- | --- | --- |
>     | ITDD | 9.71 | 0.66 | 10.90 |
>     | T5-S+LAPDOG | 14.62 | 3.23 | 14.44 |
>     | T5-B+LAPDOG | 16.08 | 3.53 | 15.33 |
>     | T5-L+LAPDOG | 17.11 | 3.56 | 15.64 |
>
> ******************Reference******************
>
> [1] Zekang Li, Cheng Niu, Fandong Meng, Yang Feng, Qian Li, and Jie Zhou. Incremental transformer with deliberation decoder for document grounded conversations. In Proceedings of the 57th Annual Meeting of the Association for Computational Linguistics, pp. 12–21, 2019.
>
> [2] Izacard, Gautier, Mathilde Caron, Lucas Hosseini, Sebastian Riedel, Piotr Bojanowski, Armand Joulin and Edouard Grave. “Unsupervised Dense Information Retrieval with Contrastive Learning.” Trans. Mach. Learn. Res. 2022.

---

### Meta-Review · Area_Chair_9u1t · 2023-09-19

**Recommendation:** 4

**Metareview:**

This paper focuses on personalized dialogue generation by leveraging external story documents. In particular, a retriever and a generator are jointly trained, which eliminates the need for annotated retrieval datasets. The retriever is utilized to extract top-k relevant stories.

In general, the reviewers agree that the paper is well-motivated and the proposed model is well-designed. However, reviewers have pointed out that more recent cutting-edge baselines and traditional knowledge-grounding baselines should be included. Reviewer d41o also pointed out that the model architecture should be described in more detail and the paper has some minor formatting issues to fix.

---

### Decision · Program_Chairs · 2023-10-07

**Decision:**

Accept-Main

**Comment:**

This paper focuses on personalized dialogue generation by leveraging external story documents. In particular, a retriever and a generator are jointly trained, which eliminates the need for annotated retrieval datasets. The retriever is utilized to extract top-k relevant stories.

In general, the reviewers agree that the paper is well-motivated and the proposed model is well-designed. However, reviewers have pointed out that more recent cutting-edge baselines and traditional knowledge-grounding baselines should be included. Reviewer d41o also pointed out that the model architecture should be described in more detail and the paper has some minor formatting issues to fix.